# Harnessing NK Cell Checkpoint-Modulating Immunotherapies

**DOI:** 10.3390/cancers12071807

**Published:** 2020-07-06

**Authors:** Sachin Kumar Singh Chauhan, Ulrike Koehl, Stephan Kloess

**Affiliations:** 1Institute of cellular therapeutics, Hannover Medical School, 30625 Hannover, Germany; Koehl.Ulrike@mh-hannover.de (U.K.); Kloess.Stephan@mh-hannover.de (S.K.); 2Fraunhofer Institute for Cell Therapy and Immunology, 04103 Leipzig, Germany; 3Institute of Clinical Immunology, University of Leipzig, 04103 Leipzig, Germany

**Keywords:** Natural killer cell (NK), cancer immunotherapy (CI), immune checkpoint costimulatory, immune therapeutics, modulatory immune checkpoint, inhibitory immune checkpoint

## Abstract

During the host immune response, the precise balance of the immune system, regulated by immune checkpoint, is required to avoid infection and cancer. These immune checkpoints are the mainstream regulator of the immune response and are crucial for self-tolerance. During the last decade, various new immune checkpoint molecules have been studied, providing an attractive path to evaluate their potential role as targets for effective therapeutic interventions. Checkpoint inhibitors have mainly been explored in T cells until now, but natural killer (NK) cells are a newly emerging target for the determination of checkpoint molecules. Simultaneously, an increasing number of therapeutic dimensions have been explored, including modulatory and inhibitory checkpoint molecules, either causing dysfunction or promoting effector functions. Furthermore, the combination of the immune checkpoint with other NK cell-based therapeutic strategies could also strengthen its efficacy as an antitumor therapy. In this review, we have undertaken a comprehensive review of the literature to date regarding underlying mechanisms of modulatory and inhibitory checkpoint molecules.

## 1. Introduction

Throughout evolution, innate immunity evolves earlier than adaptive immunity and is decisive for its exceptional line of defence in host immunity [1]. In the 1960s, a group of researchers aimed to investigate the T-cell mediated immune response, but accidentally discovered that there are naturally occurring cytotoxic lymphocytes other than T cells with antitumor properties [2]. Over the next several decades, researchers characterized this unexplored arm of the innate immune system known today as “Natural Killer (NK) Cells”.

NK cells, as their name indicates, recognize malignant cells and perform natural cytotoxic functions along with the release of chemokines and cytokines, which further impact NK effector functions [3,4,5,6]. NK cells respond to germline-encoded markers of transformation present on the surface of cancer cells and regulate this recognition function via their activating and inhibitory receptors [7,8]. The first route of this cascade includes the expression of inhibitory receptors, which include three germ-line-encoded cell surface proteins: killer immunoglobulin-like receptors (KIRs), leukocyte immunoglobulin-like receptors, and CD94/NKG2A receptor belonging to c-type lectin superfamily (Figure 1) [9,10]. On the other hand, NK cells express a balance of germ-line-encoded activation receptors, which consist of natural cytotoxicity receptors (NCRs) such as NKp80, NKp46, NKp44, NKp30, and others; the c-type lectins, a superfamily of proteins that recognize a broad range of repertoire of ligands, such as NKG2D, NKG2C/CD94, 2B4, and NTB-A, and the CD16 (FcγRIIIa), a type I transmembrane receptor containing two extracellular Ig-like domains, among others (Figure 2) [11]. 

Under malignant conditions, such as healthy cells transforming into cancer cells, stress ligands upregulate and cognately bind and activate NK receptors (Figure 2A). When activation signals dominate over inhibitory signals, NK cells activate protein tyrosine kinase (PTK)-dependent pathways and propagate the NK cell effector signalling pathways [12]. NK cells target malignant cells through an array of mechanisms. Despite these mechanisms, the malignant cells can evolve various escape routes, such as downregulation of adhesion molecules, a cognate ligand for activating receptors, or upregulation of major histocompatibility complex (MHC) molecules and secretion ofsome immunosuppressive cytokines such asinterleukin-10 (IL-10), transforming growth factor-β (TGF-β), and indoleamine 2,3-dioxygenase (IDO) [13,14,15].

## 2. NK Cells as a Potential Therapeutic Strategy

NK cells can eliminate both allogeneic and autologous malignant cells in vitro [16]. NK cells hold great potential, and understanding its mechanism of action or unravelling its exhausting mechanisms can play a crucial role in designing NK cell-based cellular therapy. 

### 2.1. NK Modulatory Mechanisms

NK cells propagate their immune response through a complex cascade of receptors, and various strategies of NK cell activation are under consideration, including antibody cross-linking, soluble ligands, and monoclonal antibodies, such as targeting NKG2D [17] and NKp30 [18], stimulated dimierized soluble cytokine Interferon-gamma (IFN-γ) release by targeting 2B4 [19], and some costimulatory receptors such as CD137 [20]. Moreover, NK cells can mediate immune defence and actively lyse antibody-tagged target cells through engagement of the CD16A receptor and KIR detection of MHC class I ligand (HLA-I) (Figure 2A,B) [21]. Thus, targeting these activating receptors will promote NK cell effector and cytotoxic function and deplete target malignant cells.

In the tumor microenvironment (TME), an early phase of CD4+T cell (also known as T helper cells) activation, induces IL-2 secretion, which leads to NK cell proliferation and enhances the cytotoxic activity of NK cells (Figure 2C) [22,23]. Mature dendritic cells (mDCs) and macrophages (M) are known as antigen-presenting cells (APC) that process and present tumor antigens to NK cells via MHC molecules [24,25]. Macrophages and mDCs release IL-23, IL-15, IL-12, IL-18, IL-27, and tumor necrosis factor-alpha (TNF-α) cytokines, which induces the maturation and activation of cytotoxic NK cells (Figure 2C) [26,27,28,29]. During tumour growth, various intrinsic cellular mechanisms induce expression of several activating ligands, a direct link between cellular transformations, apoptosis, and surveillance by the immune system [30]. For instance, the DNA damage response (DDR) induces the expression of NKG2D ligands, such as MHC class I chain-related protein A and B (MIC A/B), and upregulates the DNAX accessory molecule-1 (DNAM-1) receptor–CD155 ligand pairing [31]. 

### 2.2. NK-Suppressive Mechanisms

NK cells are significant lymphocytes, and their dysfunction has a fundamental impact on disease progression and increases tumor incidence. Tumors evade the immune response by blocking activation pathways (Figure 1). They either release soluble factors such as IL-6 and TGF-β, which downregulates several activating receptors [32,33] (Figure 1A), or interact with inhibitory receptors, such as KIRs [34], through their ligand in order to escape from the immune response. However, several studies have observed the shedding of MIC A/B (NKG2D ligand) and CD155 (also known as poliovirus receptor (PVR)) (DNAM-1 ligand) from tumour cells (Figure 1A), which prevents NK cell activation and leads to NK cell hyporesponsiveness [35,36,37,38]. Moreover, TME creates hypoxia and releases exosomes that downregulate NK cell function (Figure 1B) [39]. 

Moreover, TME also releases TGF-β and IFN-γ, and represses MIC A/B expression, which results in downmodulation of NKG2D expression, suppressing the cytotoxic ability of NK cells [40,41]. Proinflammatory or negatively regulated immune cells, myeloid-derived suppressor cells (MDSCs), dendritic cells, tumor-associated macrophages (TAMs), and regulatory T cells (Tregs) target NK cells and dampen NK cell activity through secretion of various suppressive cytokines (Figure 1C). MDSCs are capable of downregulating NK cell function through the release of TGF-β1 and IL-1β [42,43], whereas TAMs release TGF-β [44]. Moreover, Tregs can starve NK cells ofIL-2, seizing its growth, and tumor-associated fibroblasts generate Indoleamine 2, 3-dioxygenase (IDO) and Prostaglandin E2 (PGE2), which may block essential downstream NK cell activation signaling [45,46]. Similarly, DCs mediate inhibition of NK cells and require increased Signal transducer and activator of transcription 3 phosphorylation (pSTAT3) [47]. Indeed, NK cells recognize target cells without any prior activation, but malignant cells are capable of generating various ways to escape NK cell recognition and diminish their activity altogether. Subsequently, there is a new field of checkpoint receptors that regulate NK proliferation, activation, and inhibition, with growing attention towards their involvement as anticancer therapies. The therapeutic implications of the immune checkpoint molecules began by targeting inhibitory checkpoint molecules as a way of debulking tumors, but immune checkpoint costimulation has been highlighted as a target for anticancer therapies.

## 3. Fundamental Mechanisms of Immune Checkpoints

Immune checkpoints are proteins that regulate immune responses [48]. NK cells hold great potential, and unravelling these exhausting mechanisms will help play a crucial role in immunotherapy development. The concept of the immune checkpoint comes from the first proposal of anti-CTLA-4 (anti-cytotoxic T-lymphocyte-associated protein-4) as cancer immunotherapeutic target antigen in the late 1990s [49,50]. Further studies have revealed its dual functionality as both a stimulatory checkpoint and an inhibitory checkpoint [51]. Here, we discuss the recent advances in costimulator and inhibitor checkpoint molecules.

### 3.1. Immune Checkpoint Costimulators

NK cells propagate immune response through a complex cascade of receptors, and various strategies are under consideration, including the interaction of activating receptors to their cognate ligands, as well as dominant expression of activating receptors over inhibitory receptors. Recently, targeting immune checkpoints with a specific inhibitor antibody has revolutionized cancer treatment [52,53,54]. The application of immune checkpoint modification in the immunotherapy development has been explored in T cells, but checkpoint protein expression and functional significance in NK cells isless explored. In this review, we discuss the current knowledge of the expression of immune checkpoint protein of NK cells, and how they can modulate NK cell functions.

Several studies have identified costimulatory checkpoint molecules, including OX40 [55], CD357 [56], and 4-1BB [57,58] (Figure 3A). These costimulatory checkpoints on NK cells are bound by their cognate ligands from antigen-presenting cells (APCs) or tumor cells to convey proliferation, activation, or modulatory signals. 

4-1BB (CD137 or tumor necrosis factor receptor superfamily 9/TNFSF9), is a costimulatory receptor expressed on activated immune cells, such as activated T cells and NK cells [57,59], and it interacts with its cognate ligand 4-1BBL, which is present on professional antigen-presenting cells (APCs), such as dendritic cells and monocytes/macrophages [60,61]. 4-1BB/4-1BBL crosslinking can induce either costimulatory signals, leading to proliferation, activation, and IL-2 production, or induce apoptosis ofT cells in order to deplete exhausted T cells [62,63,64,65]. Chen and colleagues intensively studied 4-1BB expression and its role in eradicating 4-1BBL-positive tumor cells in a murine model. They concluded that IL-2-activated NK 1.1 murine NK cells upregulate 4-1BB expression compared to non-activated NK cells [66]. Chen et al. also determined the direct impact of IL-2 and IL-15 cytokine exposure on the 4-1BB expression of murine NK cells, and showed that crosslinking of CD137 monoclonal antibody (mAb) leads to high levels of IFN-γ secretion from NK cells [67]. Recently, human NK cells showed upregulation of the expression of 4-1BB in the presence of IL-2 and IL-15, but neither degranulation nor cytotoxicity increase demonstrated in human CD137+ NK cells [57]. Bidirectional signalling through CD137/CD137L is critical in modulating immunomodulatory functions, including the production of proinflammatory cytokines and chemokines in APCs, which attracts neutrophils. Activation of T and NK cells can influence other immunomodulatory immune cells, and CD137/CD137L signalling can also regulate functions of dendritic cells and Tregs [57,59,68,69,70,71,72]. In preclinical studies, the synergistic approach of anti-CD20 mAb and anti-CD137 mAb induced an effective NK-mediated antibody-dependent cellular cytotoxicity (ADCC) against lymphoma malignant cells [73]. There has been significant progress made in recent years in exposing the expression, function, and therapeutic potential of 4-1BB. Targeting 4-1BB with an agonistic mAb revealed potent antitumour effects in murine tumor models [74]. Currently, a combinatorial approach to synergistically combine 4-1BB therapy with approved tumor-targeted mAbs, such as a combination of Rituximab (anti-CD20 mAbs) and an agonistic 4-1BB antibody, which promotes cytokine production and enhanced tumor clearance [67,75]. Targeting of CD137 augmented the therapeutic efficacy of direct-targeting mAbs, such as Rituximab (anti-CD20) [76], Trastuzumab (anti-HER2/ human epidermal growth factor receptor 2) [77], and Cetuximab (anti-epidermal growth factor receptor (anti-EGFR)) [78], in hematological and solid tumor models. Urelumab (BMS-663513), a human IgG4 mAb was the first 4-1BB-targeting antibody to enter the clinic [79]. Initially, Urelumab showed moderate dose-dependent toxicity in phase 1 and 2 results, but later was included in trials with Rituximab, Cetuximab, Elotuzumab, and Nivolumab (NCT01471210, NCT01775631, NCT02110082, NCT02252263, and NCT02253992) (Table 1). Pfizer, New York (United States) developed a humanized IgG2 mAb agonist of the T cell costimulatory receptor 4-1BB/CD137 (Utomilumab; PF-05082566) that activates 4-1BB and blocks 4-1BBL endogenous binding. Initial clinical data with Utomilumab has shown superior safety and efficacy compared to Urelumab [80]. In 2016, Utomilumab, in combination with the humanized, (programmed cell death protein-1/PD-1-blocking IgG4 mAb Pembrolizumab, was evaluated in a phase 1b trial and demonstrated clinical benefit with two complete responses and four partial responses out of 23 patients with advanced solid tumor, and showed no significant toxicity [81]. Despite the broad expression of 4-1BB/4-1BBL and its distinct involvement in immune dynamics, the therapeutic efficacy relies directly on activated T cells and indirectly on NK cells, and can be further modified using a synergistic approach. 

Another immune checkpoint costimulatory molecule, OX40 (also known as TNFRSF4 or CD134), is a transmembrane glycoprotein and a member of the tumor necrosis factor (TNF) receptor superfamily [82,83]. CD134 is recognized as a costimulatory receptor, is transiently expressed by T cells, and is essential for regulating T cell differentiation and survival [84]. The ligand for OX40 (OX40L or CD252) belongs to the TNF superfamily and is mainly expressed by APCs such as dendritic cells, macrophages, and activated B cells, as well as T cells [85,86,87]. Among other checkpoint costimulatory molecules, OX40 is a promising therapeutic modality for inflammatory and autoimmune diseases, as it can induce antitumor immunity through effector T cell and NK cells. Preliminary studies have shown that OX40–OX40L interaction between T cells and APCs modulates T cell function [88,89,90] and is decisive for the generation of memory T cells [91,92]. Besides, the combination of cytokines IL-2, IL-15, and IL-18 can induce OX40L expression on NK cells following stimulation through activating receptors using anti-CD16 or anti-NKG2Dantibodies. Further, activated NK cells costimulate autologous CD4+ T cell proliferation and IFN-γ production via OX40–OX40L interaction [93]. This cross-talk of human NK cells and T cells demonstrates its role as a molecule relevant for linking innate and adaptive immune cells. 

In contrast to the wealth of data on T cells, the characteristics of OX40/OX40L in NK cells are poorly understood. The most recent data showed low-level expression of CD134 receptor in NK cells [94]. In a preclinical study on mice bearing established subcutaneous B16 melanoma, the interaction of CD134 on NK cells with CD134L present on dendritic cells resulted in the release of IFN-γ and induced T cell cross-priming against multiple tumor antigens [95]. In another preclinical study, human NK cells showed transiently upregulated CD134 in coculture with Rituximab-opsonized Ramos B cell lymphoma cells, and after stimulation with agonistic anti-CD134 mAb, demonstrated enhanced cytotoxicity and cytokine release in human NK cells [96]. Gene expression profiling of NK cells showed a 4-fold increases in 4-1BBL expression upon activation of NK cells [97]. Regarding other checkpoint costimulatory molecules, OX40 is the furthest along in clinical development. Despite the obscure role of OX40 in NK cells, OX40-targeting drugs utilized in initial trials and preliminary data showed satisfactory result and signs of clinical activity. In a phase 1 clinical trial, 9B12 (an OX40-targeting agent) showed good tolerance and moderate toxicity in patients with solid metastatic malignancies [98]. Currently, the outcome from a clinical trial is its safety using OX40/OX40L-targeted therapy, which indicates the requirement of further exploration. In a translation research study, a murine anti-OX40 antibody (9B12) directed against the extracellular domains of human OX40 (CD134) was tested in a phase I clinical trial, and was found to be well-tolerated and to promote both humoral and cellular immunity in patients with cancer [98]. Later, humanized monoclonal antibodies against OX40 (GSK3174998) was developed by GlaxoSmithKline PLC, Brontford (United Kingdom) in partnership with Merck & Co., New Jersey (United States) designed to bind to OX40 on activated T cells, and is currently under investigation for use alone or in combination with Pembrolizumab (anti-neoplastic agents) in advanced solid tumor patients (NCT02528357) (Table 1). 

Another modulator, GITR (glucocorticoid-induced TNFR family-related gene), a member of the TNFR superfamily (TNFRSF), was initially cloned in a glucocorticoid-treated hybridoma T cell line [99]. It has high similarity in its cytoplasmic region toother TNFRSF members [100], namely 4-1BB, CD27, OX40, and CD40, which all employ costimulatory activity [101,102]. Preliminary studies showed that Tregs express GITR and engagement with the soluble form of GITR ligand, thus abrogating the suppressive function of Tregs [103]. Later, studies also confirmed its presence on helper and cytotoxic T cells [104], macrophages, and NK cells [56,105], whereas the cognate ligand (GITRL) is expressed on antigen-presenting cells, such as dendritic cells [106]. In the case of NK cells, there is constitutively low expression of GITR, but is upregulated upon activation by both toll-like receptors (TLRs) ligand and NK cell growth factor, IL-15 [107], and binds to its respective ligand, GITRL (a type II transmembrane protein) [108]. Upon interaction, the TNFR ligand is expressed in antigen-presenting cells and forms trimers for crosslinking the receptors. Similar to OX40, GITR modulates immune cell activation through costimulation and increases the proliferation, activation, and cytokine production of cytotoxic T cells, but its role in NK cells remains controversial [109]. Hanabuchi et al. [110] demonstrated that GITR–GITRL interactions enhanced both NK cell cytotoxic activity and IFN-γ production in NK cells. However, another study reported that GITRL-positive tumor cells downregulate NK cell cytotoxicity and inhibits IFN-γ release [56]. Later, GITR signalling mechanisms were investigated, and it was concluded that GITR serves as a negative regulator of NK cell activation by enhancing NK cell apoptosis [107]. Conclusively, the interactions between NK cells with GITRL-expressing cells and the presence of reverse signalling indicate the homeostasis of NK cells during inflammation, which provides strong evidence for a much broader role than initially recognized. Moreover, its antitumor activity and expression in activated NK cells should be studied further. Initially, agonistic GITR antibodies can attenuate the suppressive function of Tregs, as well as stimulate effector T cells to overcome Tregs-mediated suppression, which makes it a promising target for clinical development [111,112,113]. Based on the preclinical activity of the agonist anti-GITR antibodies, Merghoub et al [114] investigated the first human phase 1 trial of GITR agonism with the anti-GITR antibody TRX518 (NCT01239134). TRX518, a humanized Fc-dysfunctional aglycosylated anti-human GITR mAb, induces GITR signalling. The phase I clinical trial showed preliminary affirmative response from the patients due to depletion of GITR-positive Tregs [114]. 

Thus, we postulate that NK cells mediate a protection mechanism against various stressed cells due to upregulation of the cellular stress response, which triggers the stress-induced ligand and activates the effector cells. Thus, formulating a NK cell-based immunotherapy that can concurrently engage these cellular stress responses along with immune checkpoint costimulator-based therapy will lead to a more active, targeted, and safe approach of therapeutic development. Immune checkpoint costimulators such as OX40, CD137, and GITR are expressed mostly following stimulation the lymphocytes, which makes them safe to target with agonistic antibodies [115]. 

### 3.2. Immune Checkpoint Inhibitors

The dysfunctional status of NK cells can link to exhaustion, anergy, and senescence. Immune checkpoint inhibitors, such as T cell immunoglobulin and immunoreceptor tyrosine-based inhibitory motif domains (TIGIT) and CD96, represents a type I transmembrane glycoprotein belonging to the immunoglobulin superfamily [116,117], cytotoxic T-lymphocyte-associated protein 4 (CTLA-4) [118], KIRs [34], lymphocyte-activation gene-3 (LAG-3) [119], PD-1 [120], T cell immunoglobulin and mucin-domain containing-3 (TIM-3) [121], and sialic acid-binding immunoglobulin-type lectins (SIGLECs) [122] have shown remarkable efficiency in the development of immunotherapy (Figure 3B). However, knowledge of the expression of various immune checkpoints in NK cells and dissecting the role of these inhibitory mechanisms in NK cells is essential for the full interpretation of the mode of action of immune-checkpoint-based therapy in the treatment of cancers. 

NK cells prevent healthy cell targeting through MHC-I specific inhibitory receptors, such as KIRs and CD94-NKG2A, through the immunoreceptor tyrosine-based inhibition motif (ITIM), which recognizes the human leukocyte antigen-E (HLA-E) ligand on healthy cells and suppress signalling activation [123]. However, tumor cells induce expression of the non-classical MHC class I molecule, the HLA-E ligand, in order to escape from NK cell-mediated action [124]. NKG2 lectin-like receptor (also known as CD159) is expressed as a heterodimer with CD94 on NK cells and specifically recognizes HLA-E. It has been reported that HLA-E is an effective suppressive factor, and its expression on tumor cells negatively correlates with patients’ overall survival [125,126]. Upon engagement to its cognate ligand, CD94/NKG2A delivers inhibitory signals [127,128,129]. Tumor-infiltrating NK cells express a high level of NKG2A, and the frequency of NKG2A+NK cells correlates with the secretion of IL-15 and TGF-β in solid tumor [130,131,132]. The blocking of NKG2A-HLA-E binding greatly enhances antitumor properties of tumor-infiltrating NK cells as confirmed both in vitro and in vivo [133,134]. In vitro and in vivo studies have shown the application of humanized anti-NKG2A antibodies against malignancies to be safe and effective [134,135]. It is wise to block the CD94/NKG2A receptor in order to avoid immune escape as a therapeutic strategy. Hence, a blocking antibody against CD94/NKG2A (IPH2201-Monalizumab) developed by Innate Pharma, Marseille (France) is currently under trial in phase I/II clinical trials as a monotherapy (NCT02459301) [136]. Several studies have reported that PD-1 is coexpressed along with NKG2A in tumor-infiltrating NK cells; therefore, a combination of Monalizumab and Durvalumab is currently under trial and has shown clinical efficacy [132]. Preliminary results have determined that combined blocking of the NKG2A/HLA-E and PD-1/PD-L1 pathways with blocking antibodies has promising efficacy [132,137]. Furthermore, combination of Monalizumab with other compounds such as the anti-EGFR mAb (Cetuximab) in patients with recurrent or metastatic solid tumor (NCT02643550) and inhibitor of Bruton tyrosine kinase (ibrutinib) in relapsed/refractory CLL (NCT02557516) are also under investigation. Overall, NKG2A blockade shows a promising therapeutic approach, and its combination with other checkpoint molecules is the way forward and requires further exploration. 

The KIRs (also known as CD158) are polymorphic receptors that recognize a specific HLA class I allotype (HLA-A, -B, or -C) as a cognate ligand [138]. Upon interaction with their associated ligands, KIRs execute an inhibitory signal that prevents NK cell activation towards MHC-I positive, healthy autologous cells [139,140,141]. There are a total of 17 *KIR genes* with diverse allelic combinations [142,143,144]. The clinical relevance of KIR inhibition has been reported in allogeneic haploidentical stem cell transplantation (HSCT) in AML patients from KIR ligand-mismatched donors. Activation of NK cells and eradication of residual leukaemia was reported [145,146,147]. In a clinical setting, infusion of KIR ligand-mismatched allogenic NK cells to advanced multiple myeloma patients, followed by HSCT, showed promising results along with no graft-versus-host disease [148]. Accumulating evidence indicates the benefits of pharmacological exploitation of combining KIR blockade therapy with another immunotherapy. For instance, the blockade of KIR releases the inhibition breaks and assist in Rituximab-dependent NK cell-mediated ADCC [149]. Alternatively, a combination of anti-KIR antibody (IPH 2101) and an immunomodulatory drug, i.e., Lenalidomide, in relapsed/refractory multiple myeloma patients, could be of interest in treatment [150]. Moreover, Lirilumab (IPH2102), a human IgG4 monoclonal antibody (mAb), has been evaluated for safety in various cancer patients, and a phase 1 trial confirmed its safety and blockade of KIR [151]. Furthermore, Lacutamab (also termed as IPH4102), a first-in-class humanized monoclonal antibody targeting KIR3DL2, has also been under evaluation in clinical trials, and was confirmed as a safe therapy against T cell lymphoma [152]. In clinical trials, there are limited side effects of KIR blockade alone, with slight efficacy [153]. However, the combination of PD-1 and KIR blockade, or CTLA-4 and KIR blockade, has shown increased response in chemotreated advanced head and neck cancer patients (NCT01714739). Both Lirilumab and Monalizumab (anti-NKG2A) are currently undergoing phase I/II clinical trials as monotherapies or in combination across a series of hematologic and solid cancers (Clinical trial.gov: Lirilumab: NCT02399917, NCT02599649, NCT02252263, NCT02481297, NCT01687387, NCT01714739, NCT01592370, NCT01750580; Monalizumab: NCT02921685, NCT02643550) (Table 1). The results of various clinical trials have reported that treatment with the anti-KIR antibody can induce an antitumor immune response in cancer patients [154]. 

TIGIT and CD96 are inhibitory checkpoint molecules from the same immunoglobulin superfamily, and are expressed on NK and T cells [116,155]. CD96 has lower binding affinity for the ligand CD155 compared to TIGIT, whereas DNAM-1 (CD226), an activating receptor, also competes with TIGIT and CD96 in binding to CD155 [156,157,158]. CD155 is a transmembrane glycoprotein, also known as PVR, that is highly expressed in many tumor cell lines and primary malignancies [159,160]. Various cancers have shown upregulation of CD155, which may bind to TIGIT and CD96 in order to evade NK cell-mediated antitumor immunity by eliciting NK cell inhibition, including suppression of granule polarization and IFN-γ production [161,162,163,164]. TIGIT was shown to compete for binding to its cognate ligand with higher affinity than DNAM-1 [164] and downmodulates the NK cell-effector function, whereas CD96 dampens IFN-γ production [158], which can be reversed by the disruption of interactions of TIGIT with its ligands [164]. Preclinical studies support the idea of blockading TIGIT/CD96 checkpoints to activate further NK cell-mediated antitumor immunity [157]. Patients with higher TIGIT expression in the bone marrow (BM) experience a graft-vs.-leukemia (GVL) effect and GVHD after HSCT in AML patients to control NK cell activation and proliferation. These observations conclude that TIGIT could be a prognostic predictor following HSCT and can be targeted as a potent immunotherapeutic modality in AML patients [165]. Recently, increased emphasis has placed on the combination of checkpoint inhibitors in order to produce higher efficacy. Since TIGIT acts synergistically with both TIM-3 and PD-1 to weaken the antitumor immune responses [166], a phase I trial has been evaluating a human anti-TIGIT mAb (MTIG7192A, RG6058) in combination with anti-PD-1 therapy in various solid tumors (NCT03563716). On the other hand, the preclinical study showed that CD96 mediates immune response, and blocking of CD96 promotes the release of IFN-γ and an enhanced antitumor response [167]. In a clinical trial, a pharmaceutical cocktail containing anti-TIGIT (Etigilimab) and an anti-PD-1 (Nivolumab) antibody was used to eradicate cancer and restore immune response, and the results of this trial showed the tolerability of therapy [168]. Besides, a phase III trial (named as SKYSCRAPER-01) has been started recently by Hoffmann-La Roche, Basel (Switzerland) to evaluate the efficacy and safety of Tiragolumab (anti-TIGIT, RG6058), a fully human monoclonal antibody plus Atezolizumab (anti-PD-L1), compared with placebo plus Atezolizumab, i.e., an engineered monoclonal antibody of IgG1 isotype against the protein programmed cell death-ligand 1, in participants (NCT04294810). 

Apart from these, all mammalian structures express distinct glycan structure, and Siglecs is one of them. Siglecs are immunomodulatory receptors belonging to the I-type lectin family [169]. Similar to NKG2A/CD94 and KIRs, human NK cells express inhibitory receptors, such as Siglec-7 and Siglec-9, which contain ITIM-like motifs in their cytoplasmic tail [170,171] and are known to mediate both inhibitory and apoptotic signal [172]. The hypersialylation of membrane-bound glycans and proteins is considered a hallmark of cancer, which results in the covering of malignant cells with sialic-acid-derived ligands for inhibitory Siglec receptors, and has been shown to play a significant role in immune evasion and metastasis [173,174]. Desialylation of tumor cells is an excellent fundamental idea to determine whether Siglec receptor–ligand interactions may facilitate immune evasion. Accordingly, targeting Siglecs and modulating hypersialylation have come into the limelight as potential immunotherapeutic strategies. Currently, anti-Siglec-7 and Siglec-9-blocking antibodies promote NK cell-mediated cytotoxicity against a K562 malignant cell line [171]. Malignant cells express Siglec-7 ligands to escape from the innate immune response, as well as NK cell-mediated ADCC. In vitro infusion of sialidase cuts off the sialic acid ligands and enhances HER2 positive tumor cell killing by NK cells. This result showed that desialylation of tumors make them more prone to NK cell-mediated cytotoxicity [175,176]. However, a combination of NKG2A blockade and anti-Siglec antibodies showed improved antitumor response as compared to a single treatment. These data indicate the possibility that anti-Siglec-7 and anti-Siglec-9 blocking antibodies can be a promising therapeutic option along with other immune checkpoint inhibitors. 

In 1990, Triebel and colleagues identified a novel human Ig superfamily, known as LAG-3, that is expressed on the cell surface of activated T and NK cells and is structurally similar to the CD4 receptor [177,178]. The LAG-3 receptor is expressed on other immune cell surfaces, including human tumor-infiltrating lymphocytes (TILs), Tregs, B cells, and DCs [179,180,181,182]. Usually, LAG-3 is undetectable in resting NK cells, where as it is upregulated in activated NK cells [177]. LAG-3 binds MHC-II, which is mainly expressed by APCs and malignant cells. LAG-3 also interacts with Human LSECtin (liver and lymph node sinusoidal endothelial cell C-type lectin, CLEC4G), which is widely expressed on many tumor cell surfaces [183]. The interaction of LAG-3 prevents the cytotoxic function of effector T cells and is responsible for T cell exhaustion [184,185,186]. Wherry et al. [187] have reported the coexpression of LAG-3 and PD-1 inhibitory receptors, which are involved in cytotoxic T cell exhaustion in response to viral infection, and dual blockade of these inhibitory receptors has demonstrated synergistic effects. However, additional experimental evidence suggests that LAG-3 may also play a key role in NK cell function. However, human NK cells did not show any similar results, and blocking LAG-3 did not provoke NK cell cytotoxicity [188,189]. Furthermore, a soluble recombinant LAG-3–Ig fusion protein, i.e., IMP321, has been studied as an immunological adjuvant for vaccination against cancer, and it was able to promote cytokine production of NK cells in a healthy individual and cancer patients in *ex vivo* culture conditions [190]. Relatlimab (an anti-LAG-3 mAbs), investigated either alone or in combination with PD-1 blocking antibody, has shown promising results [191,192]. Several clinical trials (NCT02658981; NCT03489369; NCT02061761; NCT01968109; NCT03005782) (Table 1) are focusing on determining the safety and efficacy of LAG-3 targeted treatment of a wide range of cancers. Hence, LAG-3 has the potential to activate NK cells, but the underlying mechanisms need further investigation.

TIM-3 is another coinhibitory receptor. It has four ligands, including galectin-9, HMGB1 (high mobility group protein B1 protein), CEACAM-1 (carcinoembryonic antigen cell adhesion molecule 1), and PtdSer (phosphatidylserine), in various malignant cells [121,155]. TIM-3, also known as hepatitis A virus cellular receptor 2, (HAVCR2) provides negative regulation to many lymphocytes [193]. The expression of TIM-3 is diverse, including distinct immune cells such as CD4^+^T cells, CD8^+^T cells, and Tregs, [194,195]. TIM-3 expression on NK cells has several aspects to it. In NK cells, TIM-3 is expressed at a low level, but a variety of unique and overlapping stimuli, including several cytokines (IL-12, IL-15, and IL-18), induce TIM-3 expression [196,197], which is considered as a marker of NK cell exhaustion [198]. The exhausted peripheral NK cells upregulate TIM-3 expression in various cancers including gastric cancers [199], lung adenocarcinoma [200], and advanced melanoma [201]. TIM-3 binding to its cognate receptors results in exhaustion of NK cells, making it a negative regulator of NK cells, but its blockade has been shown to reverse NK cell dysfunction [190,198,202]. In order to restore the NK cell activity, several antibodies for TIM-3, such as Sym023, Cobolimab, LY3321367, BGB-A425, and MBG453, in combination with several anti-PD-1/PD-L1 antibodies, are under clinical trial against various cancers [203]. Thus, targeting TIM-3 has emerged as a promising approach for checkpoint-based immunotherapy development.

CTLA-4, a coinhibitory receptor, is expressed on activated human and murine NK cells, and activates T cells and inhibits their IFN-γ release upon the engagement of its ligand B7-1, which is primarily expressed by APCs [204,205,206]. In a recent study, CD45^+^CD56^+^ human NK cells isolated from healthy donors (HD), cultured with IL-2, IL-12, and IL-18 in combination, showed synergistic high expression of the CTLA-4 receptor. IL-2 also promotes degranulation and IFN-γ production in HD [204]. This new result may indicate a relationship between CTLA-4 haploinsufficient NK cells with high susceptibility to viral infections in affected patients and heterozygous germline mutations in CTLA-4 [207,208,209]. Similarly, in vitro studies revealed that addition of IL-2, IL-12, and IL-18 to the culture conditions synergistically enhances the expression of CTLA-4, whereas TGF-β showed an adverse effect on CTLA-4 expression in murine NK cells [205]. This receptor has been studied intensively in T cells, and showed that it competes with CD28 for binding with CD80/CD86 on APCs, and that successful interaction blocks the cytotoxic function of NK cells [210,211]. CTLA-4 represents a critical checkpoint molecule as its mutation causes a complex disorder with immunodeficiency, infections, autoimmunity, and immunological disorders [207,209]. CTLA-4-blocking antibodies have been successfully trialled in murine cancer models, and are currently being implemented against human tumors [212,213,214]. Moreover, Tregs also express CTLA-4 and mediate its suppressive function [215]. In Cetuximab-treated head and neck cancer patients, CTLA-4-positive Tregs correlate with poor prognosis and suppression of NK cell cytotoxicity [216]. CTLA-4 antibody blockade increased CD4^+^ T cell proliferation, IL-2 production, and promotes proliferation and the effector function of NK cells. In addition to this, CTLA-4 blocking can relieve Tregs-mediated suppression of NK cell-cytotoxicity, and increase CD4^+^ T cell proliferation, enhance NK cell-mediated ADCC against Ipilimumab (an anti-CTLA-4 monoclonal antibody)-tagged target cells [203,217,218,219]. Accordingly, blocking or limiting CTLA-4 availability to its cognate ligands increases the availability for CD28, thus permitting the activation of potentially self-reactive intratumoral NK cells [220]. In 2010, a phase III clinical trial of Ipilimumab showed better efficacy and prolonged overall survival of metastatic melanoma patients [221], and was approved one year later by the Food and Drug Administration (FDA) for the treatment of metastatic melanoma. Recently, it was also approved for the treatment of metastatic colorectal cancer and renal cell carcinoma in combination with anti-PD-1-blocking antibody, i.e., Nivolumab [222]. Furthermore, Tremelimumab, a human CTLA-4 IgG2 antibody, has been introduced by AstraZeneca, and phase II/III clinical trials in melanoma patients showed a durable response, but did not show a significant survival benefit as compared to first-line therapy (Table 1) [223,224].

Last, but not least, programmed cell death (PD-1) is expressed on NK cells and interacts with PD-L1 or PD-L2, and their interactions have led to T cell inhibition, resulting in immune escape [225,226]. Healthy individuals express a low level of PD-1 expression, which is upregulated in peripheral and tumour-infiltrating NK cells [227]. Comparisons of PD-1^+^ NK cells and PD-1^−^NK cells have revealed the role of PD-1 in NK cell exhaustion with irregular cytotoxic and cytokine production [228]. Furthermore, a correlation between PD-1 expression and its impact on NK cell-mediated antitumor activity and disrupting PD-1 and PD-L1 interaction by blocking antibody led to partial restoration [229]. On the other hand, blocking of PD-L1 has shown to improve the NK cell functionality. In several studies, Avelumab, an anti-PD-L1 antibody-mediated ADCC toward multiple types of carcinoma cells, caused destruction of breast cancer cells and production of cytokines by Avelumab-triggered NK cells [230,231]. However, checkpoint inhibition is very complex and needs further investigation. Nivolumab (Bristol-Myers Squibb, New York (United States)) is another human PD-1 monoclonal antibody (IgG4) that has been studied in a clinical trial, and its phase I study has shown good efficacy in Non-small-cell lung carcinoma (NSCLC), melanoma, and renal cell carcinoma (Table 1) [53]. The US FDA has already approved Nivolumab for the treatment of melanoma, squamous cell lung cancer, Hodgkin’s lymphoma, and renal cell carcinoma. Other PD-1 inhibitors, such as AMP-224 and CT-011, are under clinical investigation [232,233]. 

## 4. Genetic Modification of NK Cells

A new era of cancer immunotherapy has begun with genetic modification of immune cells. Chimeric antigen receptors (CARs) are recombinant receptors, commonly formed from mAb-derived single-chain variable fragments (scFv) that provide both antigen-binding and immune-cell-activating functions [234,235]. Since various malignant cells have been shown to be able to escape from CAR-engineered cells, second and third generations of CARs including one or more endodomains for binding costimulatory molecules (i.e., CD28, 4-1BB, OX40, and 2B4) have been generated [236]. CAR binding to tumor antigens elicits downstream signalling via its intracellular domains to trigger immune cell activation, proliferation, and function. 

The first lymphocytes to be genetically modified with CAR technology were T cells, and this laid the groundwork for large-scale use of autologous CAR T cells in clinical applications to treat several types of human cancers. The FDA has recently given approval to CD19 CAR T-cell therapy to treat B cell acute lymphoid leukemia (ALL) and NHL [237]. However, the manufacturing of autologous CAR T cells is logistically challenging and expensive [238]. On the other hand, allogenic CAR-modified NK cells are attractive contenders for targeting malignant cells in the absence of a prior antigen sensitization, and extensive research efforts are ongoing to generate an off-the-shelf cellular product [239]. The CD28 motif is not naturally expressed, and their advantage in NK cell is still in discussion [240]. Currently, one clinical trial is investigating the safety and relative efficacy of CAR-CD19-CD28-zeta-2A-iCasp9-IL15-transduced cord blood-NK cells (CB-NK) in patients with relapsed/refractory CD19+ B lymphoid malignancies (NCT03056339). The advantage of CD28 costimulatory signalling is still not approved, but looks promising. 

4-1BB is a costimultory receptor expressed on activated NK cells [57]. Recently, a pilot study of redirected haploidentical NK cells was first expanded by coculture with the cell line K562-mb15-41BBL and IL-2 (NCT01974479). Similar to 4-1BB, OX40 is another checkpoint costimulatory receptor that modulates cell survival and cytokine release [93]. OX40 is often part of third-generation CAR constructs in T cells [241], but has still not been investigated in CAR constructs for primary NK cells. This technology would allow us to investigate costimulatory receptor signalling through endodomains of CAR constructs.

## 5. Concluding Remarks

Checkpoint-based cancer immunotherapy is a rapidly evolving field. Modulatory and inhibitory checkpoint functions are promising targets to enhance NK cell effector function and induce an antitumor response. However, tumor cells evade these mechanisms and dominate NK cell-based immune responses. Interaction of NK cells and tumor cells via receptor–ligand interaction is necessary to explore for the development of NK cell-based immunotherapy against cancers. Immunotherapy can enhance the antitumor immunity of the host body, and blockade of the immune checkpoint is an effective strategy to relieve the exhaustion of NK cells. Immunotherapy with checkpoint inhibitors has significantly improved the clinical outcome of various patients, but there is a need for further improvement. A transition from monotherapy to a combined synergistic approach of a distinct group of checkpoint molecules along with costimulatory molecules is on the rise in the field of immunotherapy development. Combined treatment shows higher potential in a fraction of patients that respond to medication in various cancer types. Therefore, discovering new checkpoint molecules and connecting molecules between the innate and adaptive immune system, and evaluating them for combination therapies has to be a priority in future research.

## Figures and Tables

**Figure 1 cancers-12-01807-f001:**
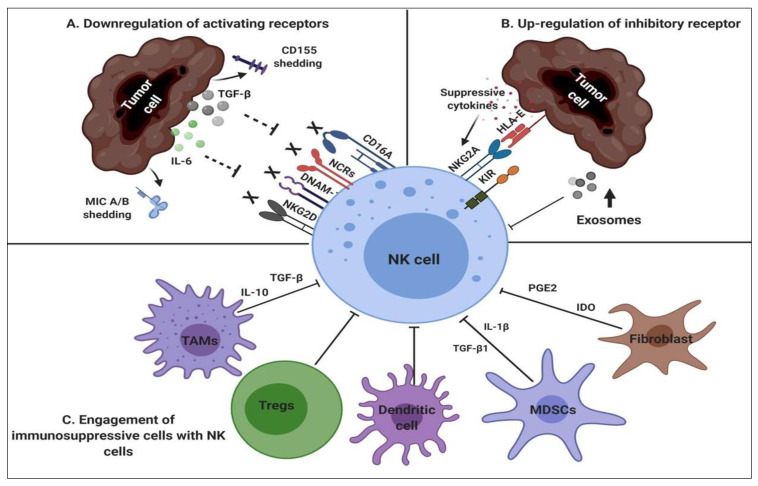
NK cell inhibitory mechanisms (**A**) Down-regulation of activating receptors on NK cells due to shedding of ligands such as CD155 or MIC A/B on tumor cells which impairs the activation of activating receptors and release of suppressive cytokines such as IL-6 and transforming growth factor-beta (TGF-ß), contribute to the formation of immunosuppressive tumor microenvironment; (**B**) Tumor cells up-regulate HLA-E ligand in order to escape from immune response which binds to NKG2A receptor and activate NK cell-inhibitory function and release of exosomes also blocks the NK cell activation; (**C**) Enrollment of immunosuppressive cells such as tumor-associated macrophages (TAMs), regulatory T cells (Tregs), Dendritic cells, myeloid derived supressor cells (MDSCs), fibroblast, suppress NK cells via suppressive cytokines.

**Figure 2 cancers-12-01807-f002:**
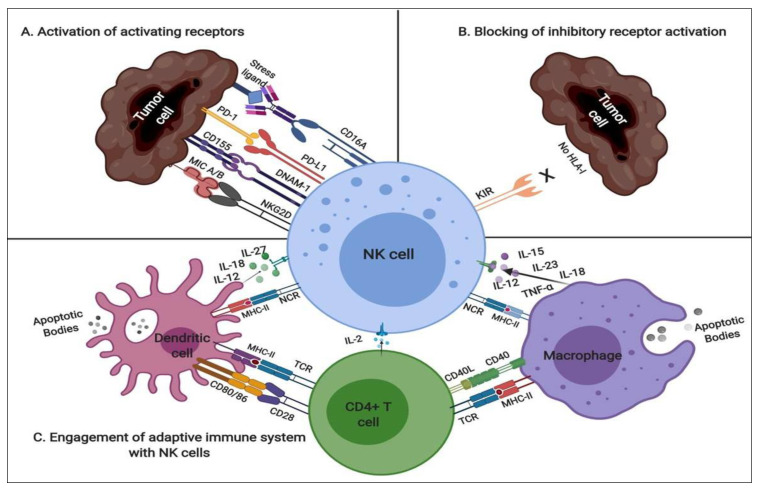
Schematic representation of NK cells-modulatory mechanism: (**A**) Mature NK cells express a range of germline-encoded activating and inhibitory receptors. The interaction of activating receptor of NK cells to their cognate ligand, either constitutively expressed or induced on tumor cells during transformation, activates the cytotoxic downstream signaling in NK cells; (**B**) As per the missing-self hypothesis, inhibitory receptor KIRs include KIR2DL1–3, KIR2DL5, and KIR3DL1–3, fails to engage with self-human leukocyte antigen (HLA) complex due to its absence and prevent the downstream inhibitory signaling; (**C**) The antigen-presenting cells (APCs) such as Macrophage and Dendritic cells process and present tumor antigen through MHC-II molecules and release distinct interleukin (IL) cytokines, encourage NK cells survival and activation; CD4+ T cell release IL-2 which is a potent inducer of NK cell proliferation and enhances cytotoxicity. (T-cell receptor (TCR), major histocompatibility complex class II molceules (MHC-II), natural cytotoxicity receptors (NCRs), tumor necrosis factor-alpha (TNF-α)).

**Figure 3 cancers-12-01807-f003:**
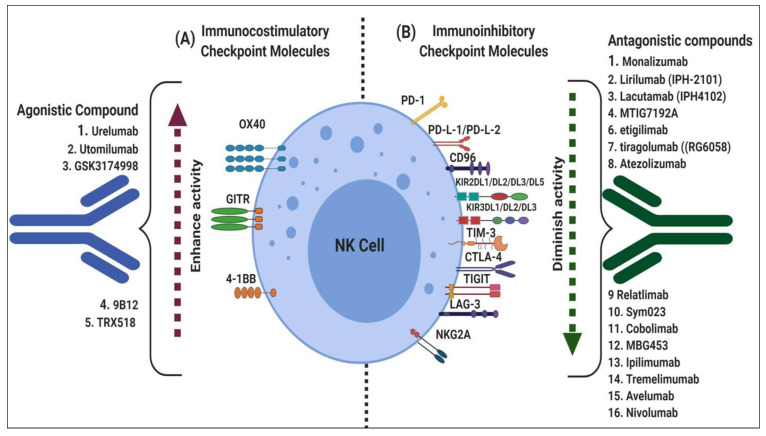
Immune-checkpoint molecules keep balance of the immune system: (**A**) Immunocostimulatory checkpoint receptors responsible for the enhanced immune cell proliferation, survival and activation and agonistic compound fully activates the receptor that it binds; (**B**) Immunoinhibitory checkpoint molecules diminish cell cytotoxic potential and prevent their antitumor functions and antagonistic compound that binds to a receptor but does not activate.

**Table 1 cancers-12-01807-t001:** Clinical trials combining checkpoint receptor based targeting with or without biologicals

Receptor/Target Antigen	Drugs /Interventions	Combinations (Drug, Biologics)	Disease	Clinical Trials, Allocation	Study Title	Participants	Status	Sponsors	Clinical Trials Identifier
4-1BB	PF-04518600 (4-1BB agonist)	Utomilumab (PF-05082566)	Neoplasms	Phase 1, Nonrandomized	Study of OX40 Agonist PF-04518600 Alone And In Combination With4-1BB Agonist PF-05082566	176	Active, not recruiting	Pfizer	NCT02315066
	CAR-T cells (PD-L1 CAR gene composed of CD137/4-1BB and CD3ζ and PD-L1 single-chain variable/ scFv fragment)	Chemotherapy	Advanced Lung Cancer	Phase 1, Not available (N/A)	CAR-T cell immunotherapy for advanced lung cancer	22	Recruiting	Sun Yat-sen University	NCT03330834
	ATOR-1017 (human mAb)	Single agent	Solid tumor, Neoplasms	Phase 1, N/A	ATOR-1017 First-in-human study	50	Recruiting	Alligator Bioscience	NCT04144842
	BMS-663513 (anti-41BB agonistic antibody)	Nivolumab, Cyclophosphamide, Fludarabine, Interleukin-2	Melanoma	Phase 1, N/A	Combining PD-1 blockade, CD137 Agonism and adoptive cell therapy for metastatic melanoma	11	Active, not recruiting	H. Lee Moffitt Cancer Center and Research Institute	NCT02652455
	PRS-343 (4-1BB/HER2 bispecific antibody)	Atezolizumab	HER2-positive Breast, Gastric, Bladder, Solid tumor	Phase 1, N/A	PRS-343 in combination with Atezolizumab in HER2-positive solid tumor	70	Recruiting	Pieris Pharmaceuticals, Inc.	NCT03650348
	Anti-CD137 (4-1BB) (BMS-663513)	Single agent	Melanoma	Phase 2, Randomized	Phase II, 2nd Line Melanoma—RAND Monotherapy	158	Completed	Bristol-Myers Squibb	NCT00612664
OX40 (CD134)	Anti-OX40 Antibody BMS 986178	TLR9 Agonist SD-101	Neoplasms	Phase 1, N/A	SD-101 and BMS-986178 in Treating Patients With Advanced or Metastatic Solid Malignancies	27	Recruiting	Ronald Levy	NCT03831295
	anti-OX40 murine monoclonal antibody, MEDI 6469	Single agent	Head and Neck Cancer	Phase 1, Nonrandomized	Anti-OX40 Antibody in Head and Neck Cancer Patients	17	Active, not recruiting	Providence Health & Services	NCT02274155
	Anti-OX40 Antibody PF-04518600	Avelumab, Binimetinib, Utomilumab	Breast Carcinoma	Phase 2, Randomized	Avelumab With Binimetinib, Utomilumab, or Anti-OX40 Antibody PF-04518600 in Treating Triple Negative Breast Cancer	150	Recruiting	Hope Rugo, MD	NCT03971409
	GSK3174998	Pembrolizumab	Neoplasms	Phase 1, Nonrandomized	GSK3174998 Alone or With Pembrolizumab in Subjects With Advanced Solid Tumors (ENGAGE-1)	142	Completed	GlaxoSmithKline	NCT02528357
	MOXR0916	Single agent	Neoplasms	Phase 1, Nonrandomized	A Study to Assess Safety and Pharmacokinetics of MOXR0916 in Participants With Locally Advanced or Metastatic Solid Tumors	174	Completed	Genentech, Inc.	NCT02219724
	anti-OX40	Radiation: Radiation, Cyclophosphamide	Prostate Cancer	Phase 1, Nonrandomized	Anti-OX40, Cyclophosphamide (CTX) and Radiation in Patients With Progressive Metastatic Prostate Cancer	13	Completed	Providence Health & Services	NCT01303705
	anti-OX40	Biological: Tetanus vaccine Biological: KLH	Advanced Cancer	Phase 1, Randomized	Phase 1 Study of anti-OX40 in Patients With Advanced Cancer	30	Completed	Providence Health & Services	NCT01644968
GITR/GITRL	Biological: TRX518	N/A	Unresectable Stage III or Stage IV Malignant Melanoma or Other Solid tumor malignancies	Phase 1, N/A	Trial of TRX518 (Anti-GITR mAb) in Stage III or IV Malignant Melanoma or Other Solid Tumors (TRX518-001)	10	Completed	Leap Therapeutics, Inc.	NCT01239134
	Anti-GITR Agonistic Monoclonal Antibody BMS-986156	Ipilimumab, Nivolumab, Radiation: Stereotactic Body, Radiation Therapy	Neoplasms & Lung carcinoma	Phase 1, 2, Nonrandomized	BMS-986156, Ipilimumab, and Nivolumab With or Without Stereotactic Body Radiation Therapy in Treating Patients With Advanced or Metastatic Lung/Chest or Liver Cancers	60	Recruiting	M.D. Anderson Cancer Center	NCT04021043
	MEDI1873	Single agent	Advanced solid tumor	Phase 1, N/A	A Study in Adult Subjects With Select Advanced Solid Tumors	40	Completed	MedImmune LLC	NCT02583165
	anti-GITR agonistic monoclonal antibody ASP1951	Pembrolizumab	Advanced solid tumor	Phase 1, Nonrandomized	A Study of ASP1951 in Subjects With Advanced Solid Tumors	435	Recruiting	Astellas Pharma Global Development, Inc.	NCT03799003
	GWN323 (Anti-GITR)	PDR001	Solid tumor	Phase 1, Nonrandomized	Phase I/Ib Study of GWN323 Alone and in Combination With PDR001 in Patients With Advanced Malignancies and Lymphomas	92	Completed	Novartis Pharmaceuticals	NCT02740270
	BMS-986156	Nivolumab	Solid tumor	Phase 1, 2, Nonrandomized	An Investigational Immuno-therapy Study of Experimental Medication BMS-986156, Given by Itself or in Combination With Nivolumab in Patients With Solid Cancers or Cancers That Have Spread.	331	Active, not recruiting	Bristol-Myers Squibb	NCT02598960
NKG2A	Monalizumab (IPH2201)	Single agent	Gynecologic Cancer	Phase 1, N/A	A Dose-Ranging Study of IPH2201 in Patients With Gynecologic Malignancies	59	Completed	Canadian Cancer Trials Group	NCT02459301
		Single agent	Hematologic Malignancies	Phase 1, N/A	Study of a Humanized Antibody Initiated 2 Months After an HLA Matched Allogenic Stem Cell Transplantation (PIRAT)	18	Recruiting	Institut Paoli-Calmettes	NCT02921685
		Cetuximab, Anti-PD-L1	Head and Neck Neoplasms	Phase 1, 2, Nonrandomized	Study of Monalizumab and Cetuximab in Patients With Recurrent or Metastatic Squamous Cell Carcinoma of the Head and Neck	140	Recruiting	Innate Pharma	NCT02643550
		Durvalumab (MEDI4736)	Advanced solid tumor	Phase 1, 2, Nonrandomized	A Study of Durvalumab (MEDI4736) and Monalizumab in Solid Tumors	383	Active, not recruiting	MedImmune LLC	NCT02671435
		Afatinib, Palbociclib, standard of care,IPH2201, Durvalumab, Niraparib, BAY1163877	Carcinoma, Squamous Cell of Head and Neck	Phase 2, Nonrandomized	Biomarker-based Study in recurrent or metastatic squamous cell carcinoma of the head and neck (UPSTREAM)	340	Recruiting	European Organisation for Research and Treatment of Cancer - EORTC	NCT03088059
		Durvalumab + Oleclumab	Stage III Non-small Cell Lung Cancer Unresectable	Phase 2, Randomized	Durvalumab Alone or in Combination With Novel Agents in Subjects With NSCLC (COAST)	300	Recruiting	MedImmune LLC	NCT03822351
KIR	Anti-KIR (1-7F9)	Single agent	Multiple Myeloma	Phase 1, N/A	An Open-label, Dose-escalation Safety and Tolerability Trial Assessing Anti-KIR (1-7F9) in Subjects With Multiple Myeloma	32	Completed	Innate Pharma	NCT00552396
	IPH2101, a human monoclonal anti-KIR antibody	Single agent	Multiple Myeloma	Phase 2, Randomized	Evaluation of Activity, Safety and Pharmacology of IPH2101 a Human Monoclonal Antibody in Patients With Multiple Myeloma (REMYKIR)	27	Completed	Innate Pharma	NCT00999830
		Single agent	Acute myeloid leukemia	Phase 1, Nonrandomized	A Safety and Tolerability Extension Trial Assessing Repeated Dosing of Anti-KIR (1-7F9) Human Monoclonal Antibody in Patients With Acute Myeloid Leukaemia	21	Completed	Innate Pharma	NCT01256073
		Single agent	Smoldering Multiple Myeloma	Phase 2, Randomized	Study on the Anti-tumor Activity, Safety and Pharmacology of IPH2101 in Patients With Smoldering Multiple Myeloma (KIRMONO)	30	Completed	Innate Pharma	NCT01222286
		Lenalidomide	Relapsed Multiple Myeloma	Phase 1, Nonrandomized	Study on the Safety, Anti-tumor Activity and Pharmacology of IPH2101 Combined With Lenalidomide in Patients With Multiple Myeloma Experiencing a First or Second Relapse (KIRIMID)	15	Completed	Innate Pharma	NCT01217203
	Lirilumab (IPH2102/BMS-986015)	Placebo	Acute myeloid leukemia	Phase 2, Randomized	Efficacy Study of Anti-KIR Monoclonal Antibody as Maintenance Treatment in Acute Myeloid Leukemia (EFFIKIR) (EFFIKIR)	152	Completed	Innate Pharma	NCT01687387
	Lirilumab (BMS-986015)	Ipilimumab	Cancer, (not-otherwise specified) NOS	Phase 1, Nonrandomized	Safety Study of BMS-986015 (Anti-KIR) in Combination With Ipilimumab in Subjects With Selected Advanced Tumor	22	Completed	Bristol-Myers Squibb	NCT01750580
		Elotuzumab, Urelumab	Multiple Myeloma	Phase 1, Randomized	A Phase I Open Label Study of the Safety and Tolerability of Elotuzumab (BMS-901608) Administered in Combination With Either Lirilumab (BMS-986015) or Urelumab (BMS-663513) in Subjects With Multiple Myeloma	44	Completed	Bristol-Myers Squibb	NCT02252263
		Rituximab	Leukemia	Phase 2, Nonrandomized	Lirilumab With Rituximab for Relapsed, Refractory or High-risk Untreated Chronic Lymphocytic Leukemia (CLL) Patients	7	Completed	M.D. Anderson Cancer Center	NCT02481297
		Nivolumab, Ipilimumab	Advanced Cancer	Phase 1, Nonrandomized	A Safety Study of Lirilumab in Combination With Nivolumab or in Combination With Nivolumab and Ipilimumab in Advanced and/or Metastatic Solid Tumors	21	Active, not recruiting	Bristol-Myers Squibb	NCT03203876
		Nivolumab, Ipilimumab	Cancer, NOS	Phase 1, 2 Randomized	A Study of an Anti-KIR Antibody Lirilumab in Combination With an Anti-PD-1 Antibody Nivolumab and Nivolumab Plus an Anti-CTLA-4 Ipilimumab Antibody in Patients With Advanced Solid Tumors	337	Completed	Bristol-Myers Squibb	NCT01714739
	IPH4102	Single agent	Cutaneous T cell Lymphoma	Phase 1, N/A	Study of IPH4102 in Patients With Relapsed/Refractory Cutaneous T cell Lymphomas (CTCL)	60	Active, not recruiting	Innate Pharma	NCT02593045
		Gemcitabine + Oxaliplatin	T cell lymphoma	Phase 2, Nonrandomized	IPH4102 Alone or in Combination With Chemotherapy in Patients With Advanced T Cell Lymphoma (TELLOMAK)	250	Recruiting	Innate Pharma	NCT03902184
TIGIT	MTIG7192A (anti-TIGIT mAb)	Atezolizumab, Carboplatin, Cisplatin, Pemetrexed, Paclitaxel, Etoposide	Advanced/Metastatic Tumors	Phase 1, Nonrandomized	Safety and Pharmacokinetics (PK) of Escalating Doses of MTIG7192A as a Single Agent and in Combination With Atezolizumab With and Without Chemotherapy in Locally Advanced or Metastatic Tumors	400	Recruiting	Genentech, Inc.	NCT02794571
		Atezolizumab, Placebo	Non-small Cell Lung Cancer	Phase 2, Randomized	A Study of MTIG7192A in Combination With Atezolizumab in Chemotherapy-Naïve Patients With Locally Advanced or Metastatic Non-Small Cell Lung Cancer	135	Active, not recruiting	Genentech, Inc.	NCT03563716
	Tiragolumab	Atezolizumab, Carboplatin, Etoposide, Placebo	Small cell lung cancer	Phase 3, Randomized	A Study of Atezolizumab Plus Carboplatin and Etoposide With or Without Tiragolumab in Patients With Untreated Extensive-Stage Small Cell Lung Cancer (SKYSCRAPER-02)	400	Recruiting	Hoffmann-La Roche	NCT04256421
		Atezolizumab, Matching Placebo	Non-small Cell Lung Cancer	Phase 3, Randomized	A Study of Tiragolumab in Combination With Atezolizumab Compared With Placebo in Combination With Atezolizumab in Patients With Previously Untreated Locally Advanced Unresectable or Metastatic PD-L1-Selected Non-Small Cell Lung Cancer (SKYSCRAPER-01)	500	Recruiting	Hoffmann-La Roche	NCT04294810
	AB154 (anti-TIGIT mAb)	AB122	Solid Tumor	Phase 1, Nonrandomized	A Study to Evaluate the Safety and Tolerability of AB154 in Participants With Advanced Malignancies	66	Recruiting	Arcus Biosciences, Inc.	NCT03628677
LAG-3	Sym022	Single agent	Metastatic Cancer, Solid Tumor, Lymphoma	Phase 1, N/A	Sym022 (Anti-LAG-3) in Patients With Advanced Solid Tumor Malignancies or Lymphomas	15	Completed	Symphogen A/S	NCT03489369
		Sym021 (anti-PD-1 mAb), Sym023 (anti-TIM-3 mAb)	Metastatic Cancer, Solid Tumor, Lymphoma	Phase 1, Nonrandomized	Sym021 Monotherapy and in Combination With Sym022 or Sym023 in Patients With Advanced Solid Tumor Malignancies or Lymphomas	102	Recruiting	Symphogen A/S	NCT03311412
	BMS-986213	Relatlimab, Nivolumab	Neoplasms	Phase 1, 2, Randomized	An Investigational Immuno-therapy Study to Assess the Safety, Tolerability and Effectiveness of Anti-LAG-3 With and Without Anti-PD-1 in the Treatment of Solid Tumors	1500	Recruiting	Bristol-Myers Squibb	NCT01968109
	BMS-986016 (Relatlimab)	BMS-936558	Hematologic Neoplasms	Phase 1, 2 Nonrandomized	Safety Study of Anti-LAG-3 in Relapsed or Refractory Hematologic Malignancies	109	Active, not recruiting	Bristol-Myers Squibb	NCT02061761
		Anti-PD-1, Anti-CD137	Glioblastoma, Gliosarcoma, Recurrent Brain Neoplasm	Phase 1, Nonrandomized	Anti-LAG-3 Alone & in Combination with Nivolumab Treating Patients with Recurrent glioblastoma multiforme (GBM) (Anti-CD137 Arm Closed 10/16/18)	63	Active, not recruiting	Sidney Kimmel Comprehensive Cancer Center at Johns Hopkins	NCT02658981
		Nivolumab	Cancer	Phase 1, Nonrandomized	Safety Study of BMS-986016 With or Without Nivolumab in Patients With Advanced Solid Tumors	45	Active, not recruiting	Bristol-Myers Squibb	NCT02966548
		Nivolumab, Carboplatin, Paclitaxel Radiation: Radiation	Gastric Cancer, Esophageal Cancer, Gastro Esophageal Cancer	Phase 1, Nonrandomized	Nivolumab or Nivolumab/Relatlimab Prior to Chemoradiation With II/III Gastro/Esophageal Cancer	25	Recruiting	Sidney Kimmel Comprehensive Cancer Center at Johns Hopkins	NCT03044613
		Drug: Nivolumab	Glioblastoma	Phase 1, N/A	Cytokine Microdialysis for Real-Time Immune Monitoring in Glioblastoma Patients Undergoing Checkpoint Blockade	25	Recruiting	National Institute of Neurological Disorders and Stroke (NINDS)	NCT03493932
		Nivolumab	Chordoma, Locally Advanced Chordoma, Metastatic Chordoma, Unresectable Chordoma	Phase 2, N/A	Nivolumab and Relatlimab in Treating Participants With Advanced Chordoma	20	Recruiting	Jonsson Comprehensive Cancer Center	NCT03623854
		Nivolumab	Melanoma	Phase 2, Randomized	Nivolumab, BMS-936558 in Combination With Relatlimab, BMS-986016 in Patients With Metastatic Melanoma Naïve to Prior Immunotherapy in the Metastatic Setting	42	Recruiting	John Kirkwood	NCT03743766
	IMP321 (eftilagimod alpha)	Placebo, Paclitaxel	Adenocarcinoma Breast Stage IV	Phase 2, Randomized	IMP321 (Eftilagimod Alpha) as Adjunctive to a Standard Chemotherapy Paclitaxel Metastatic Breast Carcinoma	241	Active, not recruiting	Immutep S.A.	NCT02614833
		Pembrolizumab	Stage III, IV Melanoma	Phase 1	Phase 1 Study of IMP321 (Eftilagimod Alpha) Adjuvant to Anti-programmed cell death protein -1 (PD-1) Therapy in Unresectable or Metastatic Melanoma	24	Completed	Immutep Australia Pty. Ltd.	NCT02676869
		Avelumab	Solid Tumors, Peritoneal Carcinomatosis	Phase 1, Nonrandomized	Feasibility and Safety of IMP321 for Advanced Stage Solid Tumors	26	Active, not recruiting	InstitutfürKlinischeKrebsforschung IKF GmbH at KrankenhausNordwest	NCT03252938
		Pembrolizumab	Non-small-cell lung carcinoma (NSCLC), Head and Neck squamous cell carcinoma (HNSCC)	Phase 2, Nonrandomized	Combination Study With Soluble LAG-3 Fusion Protein Eftilagimod Alpha (IMP321) and Pembrolizumab in Patients With Previously Untreated Unresectable or Metastatic NSCLC, or Recurrent PD-X Refractory NSCLC or With Recurrent or Metastatic HNSCC (TACTI-002)	109	Recruiting	Immutep S.A.	NCT03625323
TIM-3	Sym023 (anti-TIM-3)	Single agent	Metastatic Cancer Solid Tumor Lymphoma	Phase 1, N/A	Sym023 (Anti-TIM-3) in Patients With Advanced Solid Tumor Malignancies or Lymphomas	24	Active, not recruiting	Symphogen A/S	NCT03489343
	Drug: TSR-022 (anti-TIM-3 mAb)	TSR-042 (an anti-PD-1 antibody), TSR-033 (an anti-LAG-3 antibody)	Advanced or Metastatic Solid Tumors	Phase 1, Nonrandomized	A Phase 1 Study of TSR-022, an Anti-TIM-3 Monoclonal Antibody, in Patients With Advanced Solid Tumors (AMBER)	873	Recruiting	Tesaro, Inc.	NCT02817633
		TSR-042	Adult Primary Liver Cancer Advanced Adult Primary Liver Cancer Localized Unresectable Adult Primary Liver Cancer	Phase 2, N/A	TSR-022 (Anti-TIM-3 Antibody) and TSR-042 (Anti-PD-1 Antibody) in Patients With Liver Cancer	42	Recruiting	University of Hawaii	NCT03680508
		Dostarlimab (TSR-042) (singly)	Melanoma Stage III, IV	Phase 2, Randomized	Neoadjuvant PD-1 Inhibitor Dostarlimab (TSR-042) vs. Combination of Tim-3 Inhibitor TSR-022 and PD-1 Inhibitor Dostarlimab (TSR-042) in Melanoma	56	Recruiting	Diwakar Davar	NCT04139902
	LY3321367 (anti-TIM-3 mAb)	LY3300054	Solid Tumor	Phase 1, Nonrandomized	A Study of LY3321367 Alone or With LY3300054 in Participants With Advanced Relapsed/Refractory Solid Tumors	275	Active, not recruiting	Eli Lilly and Company	NCT03099109
	BGB-A425 (anti-TIM-3 mAb)	Tislelizumab	Locally Advanced or Metastatic Solid Tumors	Phase 1, 2, Nonrandomized	Study of BGB-A425 in Combination With Tislelizumab in Advanced Solid Tumors	162	Recruiting	BeiGene	NCT03744468
	MBG453 (anti-TIM-3 hmAb)	PDR001 (anti-PD-1), Decitabine	Advanced Malignancies	Phase 1, 2, Nonrandomized	Phase I-Ib/II Study of MBG453 as Single Agent and in Combination With PDR001 in Patients With Advanced Malignancies	269	Recruiting	Novartis Pharmaceuticals	NCT02608268
		Decitabine, PDR001	Leukemia, Myeloid Leukemia, Acute Myeloid Leukemia, Myelodysplastic Syndromes, Preleukemia, Bone Marrow Diseases, Hematologic Diseases	Phase 1, Randomized	Study of PDR001 and/or MBG453 in Combination With Decitabine in Patients With AML or High Risk MDS	235	Recruiting	Novartis Pharmaceuticals	NCT03066648
		Spartalizumab (PDR001)	Glioblastoma, Multiforme	Phase 1, N/A	Trial of Anti-Tim-3 in Combination With Anti-PD-1 and Stereotactic radiosurgery (SRS) in Recurrent GBM	15	Active, not recruiting	Sidney Kimmel Comprehensive Cancer Center at Johns Hopkins	NCT03961971
CTLA-4	AGEN1181 (anti-CTLA-4 mAb)	AGEN2034 (anti-PD-1)	Advanced Cancer	Phase 1, Randomized	Fc-Engineered Anti-CTLA-4 Monoclonal Antibody in Advanced Cancer	86	Recruiting	Agenus Inc.	NCT03860272
	Tremelimumab (anti-CTLA-4 mAb)	Durvalumab, Fulvestrant	Breast Cancer	Phase 2, Nonrandomized	Anti PD-L1 Antibody + Anti CTLA-4 Antibody in Combination With Hormone Therapy in Patients With Hormone Receptor Positive HER2-negative Recurrent or Metastatic Breast Cancer	33	Current Unknown	Kyoto Breast Cancer Research Network	NCT03430466
		Azacitidine, Durvalumab	Head and Neck Cancer	Phase 1, 2, N/A	Azacitidine, Durvalumab, and Tremelimumab in Recurrent and/or Metastatic Head and Neck Cancer Patients	59	Recruiting	Massachusetts General Hospital	NCT03019003
	Zalifrelimab (AGEN1884) (anti-CTLA-4)	Balstilimab (AGEN2034) (anti-PD-1)	Cervical Cancer	Phase 2, Randomized	Phase 2 Study of Anti-PD-1 Independently or in Combination With Anti-CTLA-4 in Second-Line Cervical Cancer	200	Recruiting	Agenus Inc.	NCT03894215
	PD-L1/CTLA4 BsAb	FOLFIRINOX (chemo drug)	Locally Advanced and Metastatic Pancreatic Cancer	Phase 1, 2, Nonrandomized	Study of Immunotherapy Combined With Chemotherapy in Locally Advanced and Metastatic Pancreatic Cancer	60	Recruiting	Changhai Hospital	NCT04324307
	AGEN1884 (anti-CTLA-4 mAb)	PD-1/PD-L-1 inhibitor	Advanced Solid Cancers Refractory to PD-1	Phase 1, 2, N/A	AGEN1884, an Anti-CTLA-4 Human Monoclonal Antibody in Subjects With Advanced or Refractory Cancer and Who Have Progressed With PD-1/PD-L1 Inhibitor as Their Most Recent Therapy	90	Recruiting	Agenus Inc.	NCT02694822
	Ipilimumab (MDX-010) (BMS-734016) (anti-CTLA-4 mAb)	Docetaxel	Prostate Cancer	Phase 2, Randomized	Comparison Study of MDX-010 (CTLA-4) Alone and Combined With Docetaxel in the Treatment of Patients With Hormone Refractory Prostate Cancer	N/A	Completed	Bristol-Myers Squibb	NCT00050596
		Single agent	Melanoma	Phase 2, Randomized	Study of Ipilimumab (MDX-010) Monotherapy in Patients With Previously Treated Unresectable Stage III or IV Melanoma	210	Completed	Bristol-Myers Squibb	NCT00289640
		REGN2810, Pembrolizumab	Non-small Cell Lung Cancer	Phase 3, Randomized	REGN2810 (Anti-PD-1 Antibody), Platinum-based Doublet Chemotherapy, and Ipilimumab (Anti-CTLA-4 Antibody) Versus Pembrolizumab Monotherapy in Patients With Lung Cancer	5	Active, not recruiting	Regeneron Pharmaceuticals	NCT03515629
		SHR-1210	Non-small Cell Lung Cancer	Phase 1, N/A	Anti-CTLA-4 Antibody Followed by Anti-PD-1 Antibody in Recurrent or Metastatic NSCLC (SEQUENCE)	10	Current Unknown	Sun Yat-sen University	NCT03527251
	AK104, a PD-1 and CTLA-4 bispecific antibody	Single agent	Advanced solid tumor	Phase 1, 2, N/A	Safety and Efficacy of AK104, a PD-1/CTLA-4 Bispecific Antibody, in Selected Advanced Solid Tumors	120	Not yet recruiting	Akeso	NCT04172454
	Tremelimumab	Olaparib	Ovarian Cancer Fallopian Tube Cancer Peritoneal Neoplasms	Phase 1, 2, N/A	PARP-inhibition and CTLA-4 Blockade in BRCA-deficient Ovarian Cancer	50	Recruiting	New Mexico Cancer Care Alliance	NCT02571725
	MGD019 (DART protein binding PD-1 and CTLA-4)	Single agent	Solid Tumor, Adult Advanced Cancer	Phase 1, N/A	MGD019 DART Protein in Unresectable/Metastatic Cancer		Recruiting	MacroGenics	NCT03761017
	CS1002 (anti-CTLA-4 mAb)	CS1003 (anti-PD-1 mAb)	Solid tumor, Adult	Phase 1, Randomized	A Study of CS1002 in Subjects With Advanced Solid Tumors	108	Recruiting	CStone Pharmaceuticals	NCT03523819
PD-1	Pembrolizumab (MK-3475) (lambrolizumab)	Single agent	Anal Cancer	Phase 2, N/A	Pembrolizumab in Refractory Metastatic Anal Cancer	32	Recruiting	Dana-Farber Cancer Institute	NCT02919969
		Radiation: RT Boost	Breast Cancer	Phase 1, 2, N/A	Breast Cancer Study of Preoperative Pembrolizumab + Radiation	60	Recruiting	Stephen Shiao	NCT03366844
	Nivolumab (anti-PD-1 mAb) (Opdivo)	Single agent	Prostate Cancer	Phase 2, Nonrandomized	Nivolumab in Patients With High-Risk Biochemically Recurrent Prostate Cancer	34	Recruiting	Beth Israel Deaconess Medical Center	NCT03637543
		DKN-01	Biliary tract cancer	Phase 2, Nonrandomized	Study of the Combination of DKN-01 and Nivolumab in Previously Treated Patients With Advanced Biliary Tract Cancer (BTC)	30	Recruiting	Massachusetts General Hospital	NCT04057365
	Atezolizumab (anti-PD-1 mAb) (MPDL3280A) (RG7446)	Single agent	Non-small Cell Lung Cancer	Phase 2, N/A	Atezolizumab in Advanced Non-small Cell Lung Cancer With Rare Histologies (CHANCE Trial) (CHANCE)	43	Recruiting	GruppoOncologicoItaliano di RicercaClinica	NCT03976518
		Single agent	Breast Cancer	Phase 2, N/A	A Study of Atezolizumab in Participants With Locally Advanced or Metastatic Urothelial Bladder Cancer (Cohort 1)	119	Active, not recruiting	Hoffmann-La Roche	NCT02951767
	Avelumab (MSB0010718C)	Single agent	Metastatic Colorectal Cancer	Phase 2, N/A	Avelumab for microsatellite instability-high (MSI-H) or POLE (i.e. Mutations in the exonuclease domain of the DNA polymerase epsilon (POLE) gene) Mutated Metastatic Colorectal Cancer	33	Active, not recruiting	Asan Medical Center	NCT03150706
		Axitinib	Cervical Cancer	Not applicable	Avelumab With Axitinib in Persistent or Recurrent Cervical Cancer After Platinum-based Chemotherapy (ALARICE)	23	Recruiting	The University of Hong Kong	NCT03826589
	Durvalumab (MEDI4736)	Single agent	Bladder Cancer	Phase 2, N/A	Efficacy of Durvalumab in Non-muscle-invasive Bladder Cancer	39	Recruiting	Hellenic Genito Urinary Cancer Group	NCT03759496
		Azacitidine, Tremelimumab	Head and Neck Cancer	Phase 1, 2, N/A	Azacitidine, Durvalumab, and Tremelimumab in Recurrent and/or Metastatic Head and Neck Cancer Patients	59	Recruiting	Massachusetts General Hospital	NCT03019003
	Cemiplimab (REGN-2810) (Libtayo)	Plerixafor	Metastatic Pancreatic Cancer	Phase 2, N/A	Plerixafor and Cemiplimab in Metastatic Pancreatic Cancer	21	Not yet recruiting	Sidney Kimmel Comprehensive Cancer Center at Johns Hopkins	NCT04177810
		REGN5678	Metastatic Castration-resistant Prostate Cancer	Phase 1, 2, Nonrandomized	Study of REGN5678 (Anti-PSMAxCD28) With Cemiplimab (Anti-PD-1) in Patients With Metastatic Castration-resistant Prostate Cancer	123	Recruiting	Regeneron Pharmaceuticals	NCT03972657

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
