# Peer review of "Harnessing NK Cell Checkpoint-Modulating Immunotherapies"

_cancers, 2020, doi:10.3390/cancers12071807_

Round 1
Reviewer 1 Report
The review by Chauhan et al. entitled “Harnessing NK Cell-Checkpoint Modulating Immunotherapies” is a very comprehensive and up-to-date summary of the evolving filed of NK-Cell therapies. There is only one major issue that needs to be addressed:
- Authors mention early clinical trials with NK-cell checkpoint inhibitors. It would increase the value of the review article, if authors included a table with ongoing trials and already finished trials (e.g. including information on: NCT trial numbers, type of drug, type of disease, Phase (I-III), in-/exclusion criteria, dosages, side effects, early efficacy).
Minor point:
- Although authors focus on checkpoint inhibitors for NK-cell based therapies, it would be nice to mention that NK-cells are also becoming a target for autologous CARs. Just a paragraph with some information on this development would strengthen the theory of NK-cell based therapies in various types of cancers.
Reviewer 2 Report
In the present review authors provide an overview of receptors that may be involved in the positive or negative regulation of NK cell functions, defining them as checkpoint receptors. They also discuss potential strategies that could modulate their function to improve anti-tumor NK cell activity.
These issues are of interest and indeed, have been often discussed in several reviews in these years. For this reason, a general consideration is that the present review should offer some more novel insights or perspectives to be more attractive. Moreover, there are concerns that the review in its present form may be suitable for publication.
Major points
1) References are often lacking or not fully appropriate.
2) There are some sentences which are not clear:
- page 2, lines 61-63 “Also, mature Dendritic cells (mDCs) and macrophages (M) are known as antigen-presenting cells (APC), which represent tumor antigen to NK cell-like effector cells.”?
- page 6 lines 218-220 “ Changes in sialic acid-like hypersialylation, xenosialylation, and sialic acid alterations associated with cancer growth and can hypersialylation targeted for eliciting anti-tumor immunity.”?
3) There are some doubts about th the way these checkpoint receptors are defined and/or described.
- a) Immune checkpoint modulators
- It is unusual defining the NKG2D activating receptor as an immune checkpoint modulator. NKG2D may be inhibited by several tumor-mediated immune-escape mechanisms (as well described by the authors) but its role it is always an activating role when it is allowed to exert its functions. Otherwise, in the same way, other main activating receptors such as NKp44 might be considered a modulatory checkpoint receptor. It has been shown that there are some NKp44 ligands (such as soluble Nidogen-1 molecule) that can inhibit NKp44 activating functions: author could add some information regarding these data and regarding potential mechanisms to overcome these inhibitory immune-escape mechanisms (Parodi M. et al. Frontiers in Immunology 2019, Gaggero S, et al. Oncoimmunology 2018, Barrow AD. et al. Frontiers in Immunology 2019). Of note, in Figure 1, authors describe the possibility that generic NCR (i.e. Natural Cytotoxicity Receptors?) may recognize MHC class II molecules. Maybe, they were thinking of paper by Niehrs et al. published last year on Nature Immunology, that indicate a subset of HLA-DP molecules as potential ligands for NKp44. However, that paper is not listed in the references and the authors do not discuss these data, while it could have been of interest.
- The other “modulatory” checkpoint receptors described are not main activating receptors and may be considered as co-receptor. Importantly, authors report that the majority of these co-receptors have been described to exert a role in T cell activation while their role in human NK cells is still not clear or at least, it appears less relevant. Thus, in reviewer’s opinion, their association with NKG2D is not fully appropriate. Authors report some data on immunotherapy protocols that involved these receptors, but they do not provide evidences that these protocols have a direct positive effect on NK cell antitumor activity.
- b) Immune checkpoint inhibitors
- In the description of TIGIT/CD96 receptors, it could have been important adding the information that they compete for the same ligands of the activating receptor DNAM-1expressed by NK cells. Moreover, it seems that sentence at page 6, line 213, should be supported by an appropriate reference.
- The first sentences that describe CTLA-4 may cause misunderstandings and the use of references is a little puzzling. As appropriately reported, CTLA-4 it has been the first well described immune checkpoint inhibitory receptor. Authors report that CTLA-4 expression is detected in several immune cells but references are lacking (page 15, line 256). Generally, human NK cells do not express CTLA-4 however, the way that authors used to describe CTLA-4 features, leads to the idea that this receptor may be easily expressed by NK cells and may play a direct role in NK cell regulation. Of note, any of the references indicated (nn. 108,109) support the idea that CTLA-4 may be expressed by Natural Killer cells or directly influencing their activity. In this context, there are also no references regarding the sentence at page 15, lines 258-259. Finally, reference 110 is very old and report the effect of IL-2 administration on peripheral blood lymphocytes, while reference n. 111 does not report data on NK cells. So it should be better explained how CTLA-4 modulation may improve NK cell function.
- Minor points
Figure legends should be improved.
Round 2
Reviewer 2 Report
The review shows a clear improvement . I appreciated the editing job done either on the references list either on the different issues to discuss.There is still doubts about some statements:
1) What is the meaning of NK -like effector cells ( What kind of cells authors were thinking of? NK-CTL, NK-T?..) described at page 3 lane 62?
2) Statements at page 5 , lanes 168-170 and the one at page 8, lanes 229-232, are not clear. In the last one, it seems a verb is lacking.
There are still some tips error and maybe some verbal forms that could be edited.
3) It seems that often in the text the space key has not been used. So a careful check all through the text should be performed, i.e.
Page 2, lane 47; page 4 lane 91;page 5 lane 119; page 5 lanes 159, 160,166 and 169;page 8 lane 228 and so on.
4) Page 7, lane 195 please edit : " ..that GITR is expressed by Tregs" or " that Tregs express GITR"; similarly, page 10, lane 314, please edit: "..and Siglec 9 are expressed by NK cells"; page 9, lane 273, please edit: "has been evaluated".
5) Page 9, lanes 264 and 267, it would be more appropriate the acronym "HSCT" rather than "alloSCT"
6) Page 10 lane 305, Nivolumab is an anti-PD1 mAb, not anti-PDL1.
